# ToG-I: Progressively Instructed Knowledge Graph-based Large Language Model Reasoning

## Abstract

Large language models (LLMs) reasoning based on knowledge graphs (KGs), by integrating structured knowledge from the KGs, provide a significant solution to alleviate the hallucination problem in complex reasoning tasks. Current techniques mainly focus on the retrieval of explicit knowledge from KGs. LLMs directly use the specific facts and relationships retrieved to construct a reasoning chain to answer the questions. However, these methods often overlook the significance of comprehending implicit knowledge when dealing with problems involving logical reasoning or ambiguous intentions. This could potentially lead to deviations in the reasoning path, hindering their applicability in real-world applications. In this paper, we propose a progressive instructed reasoning framework, ToG-I. The framework identifies core elements, discerns latent intentions, and integrates necessary commonsense reasoning by analyzing the problem from multiple perspectives and levels. Based on this, ToG-I transforms these analysis results into specific reasoning instructions, guiding the LLMs to carry out a progressive reasoning process from a global perspective. This not only ensures the accuracy of the reasoning process but also effectively avoids unnecessary consumption of reasoning resources. Extensive experiments on multiple public datasets show that ToG-I achieves state-of-the-art performance in KG reasoning tasks based on information retrieval and demonstrates superiority in knowledge-intensive tasks.

## 1 Introduction

In recent years, the capabilities of LLMs have grown rapidly, demonstrating significant superiority in a wide range of natural language processing tasks(Achiam et al., 2023; Touvron et al., 2023; GLM et al., 2024). However, the problems they present cannot be ignored. Firstly, the hallucination problem(Ji et al., 2023) is a common issue for large models. Especially when LLMs face complex scenarios that require deep understanding, they often tend to have reasoning biases, leading to output results that deviate from the actual situation. Additionally, timeliness is another major challenge faced by LLMs. Data and environments in the real world are constantly changing, and fine-tuning LLMs not only consumes significant resources but also poses the risk of catastrophic forgetting(Razdaibiedina et al., 2023). Lastly, LLMs also face significant challenges in specific domains(Wang et al., 2023a). Due to the varying knowledge systems and characteristics across different fields, it is difficult for LLMs to achieve comprehensive coverage.

The introduction of KG offers an effective solution for LLMs in addressing the aforementioned problems(Pan et al., 2024). As a structured form of knowledge representation, KG integrates entities, attributes, and relationships to construct a semantically rich complex network, thereby presenting knowledge in a clear and explicit manner. Leveraging the clear relational structure and logical framework of KG, LLM has a reliable knowledge source for reasoning, which significantly enhances its reliability and logicality when handling complex reasoning tasks. The flexible update mechanism of KG allows it to dynamically capture and reflect the latest facts and information, effectively compensating for the timeliness limitations of LLM. Furthermore, by customizing knowledge bases for specific domains, KG further improves the performance of LLM in professional fields, enhancing the model's advantages in terms of domain knowledge accuracy and specificity(Agrawal et al., 2024).

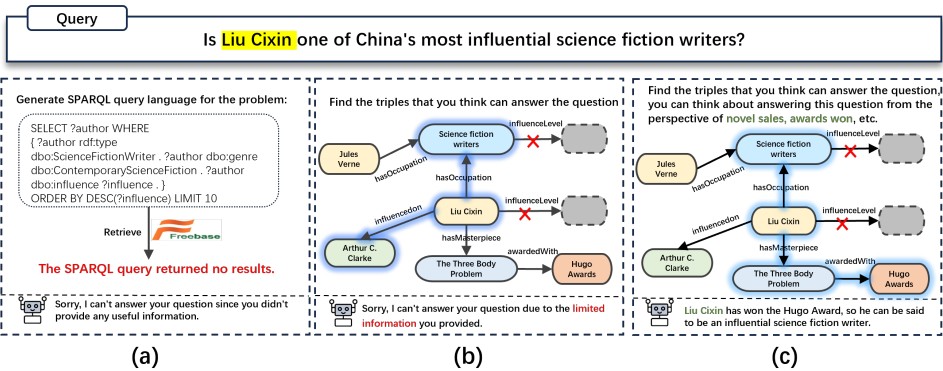

Figure 1: Different information retrieval methods in KG: (a) Semantic Parsing (b) Reasoning Path Exploration (c) Instruction-based Reasoning Path Exploration. The glowing entities or relationships represent the exploration path of the LLM.

Utilizing KG to enhance LLM's question-answering, the main purpose is to retrieve knowledge from the KG based on the question to obtain a response. One mainstream method is semantic parsing(Zhang et al., 2023; Xie et al., 2022; Ye et al., 2021; Li et al., 2023), which transforms natural language questions into the logical query language of KG through LLM. However, the effectiveness of this method largely depends on LLM's understanding of natural language, as well as the quality and completeness of KG. It is difficult to effectively align questions with KG knowledge, especially in dealing with complex, knowledge-intensive tasks (as shown in Figure 1(a)). Another more flexible approach is retrieval enhancement, which retrieves the most likely triples to answer the question through different knowledge retrieval methods, and prompts LLM to answer the question based on these evidence triples. Traditional methods rely on multiple steps such as entity recognition, relation extraction, query graph construction, and result inference to align natural language questions with structured knowledge bases(Saxena et al., 2020; Shi et al., 2021; Zhang et al., 2022). However, this multi-step processing is complex to effectively transfer in cross-domain problems. Errors accumulate gradually in the multi-hop inference process, making it difficult to handle complex problems.

Therefore, existing researchers tend to leverage the powerful language understanding capabilities of LLM in combination with KG for collaborative reasoning(Jiang et al., 2023; Guan et al., 2024; Liu et al., 2024; Wen et al., 2023). For example, Sun et al. (2024) proposed a responsible and flexible TOG algorithm, which guides LLM to perform beam search on the KG, iteratively exploring multiple reasoning paths until a path that can answer the question is found. The process includes search pruning and reasoning decisions, with the aim of selecting the most potential paths and determining whether they are sufficient to answer the question. However, this algorithm has limitations when dealing with long-distance reasoning or questions with unclear intentions. When the key evidence in the question is weakly associated with the subject entity or is far apart, LLM lacks global guidance when reasoning, making it difficult to explore the correct path from the massive relationships or entities, which may lead the reasoning process astray. As shown in Figure 1(b), for such an intentionambiguous question "Is Liu Cixin one of China's most influential science fiction writers?", LLM may choose the wrong reasoning path due to a one-sided understanding of the question, leading to the failure of the question's reasoning. In addition, although the multi-path exploration method based on the beam search algorithm alleviates this problem to some extent, the algorithm lacks flexibility when facing problems of different difficulty levels, leading to a decrease in efficiency and waste of resources.

Based on the above problems, this paper proposes a progressive instructed reasoning framework. Before reasoning, with the powerful natural language understanding capabilities of LLMs, we perform multi-angle, multi-level deep analysis of the problem, and obtain different instructive reasoning ideas for the problem from shallow to deep. At the same time, different subject entities are extracted from the reasoning ideas as the starting points for global reasoning. Then, based on instructive thinking at different levels, the large model is instructed to explore entities and relationships. As shown in Figure 1(c), with the help of LLM, we can obtain the following instructive opinions by performing multi-angle analysis and understanding: "You can search for Liu Cixin's related works and their sales rankings." "You can search whether Liu Cixin's representative works have won well-known science fiction awards (such as the Hugo Award, Nebula Award, etc.)." Under the instruction

of these strategies, even if the connections between evidence entities are relatively distant, the LLM can accurately explore the correct reasoning path from a global perspective. In addition, to increase the framework's flexibility for problems of varying difficulty, we gradually increase the range of LLM's exploration on the graph based on the depth of the instructive ideas. This dramatically reduces the number of LLM calls and reasoning time while ensuring the quality of the framework's answers to questions.

## 2 RELATED WORKS

### 2.1 LLM REASONING

In natural language processing, LLMs such as GPT, GLM, etc., can demonstrate significant capabilities in handling various tasks with simple prompts. However, standard prompting methods often only guide the model to generate direct answers, ignoring the key steps in the reasoning process. In recent years, the introduction of Chain-of-Thought (CoT) technology(Wei et al., 2022) has opened up new paths for improving the performance of LLMs in various reasoning tasks. Through specific prompting strategies, CoT technology guides the model to derive a series of intermediate steps or sub-goals before generating the final answer, enabling the model to exhibit a thinking process closer to that of humans in complex reasoning tasks. To advance the development of CoT technology, researchers have developed a variety of strategies, including automating the construction of CoT reasoning processes using the built-in knowledge bases of LLMs(Shao et al., 2023; KaShun et al., 2023), and further enhancing the application effectiveness and efficiency of CoT technology through strategies such as subproblem decomposition(Zhou et al., 2023), self-consistency(Wang et al., 2023b; Zelikman et al., 2022), building thought trees(Yao et al., 2024), and thought maps(Besta et al., 2024).

However, the knowledge of LLM itself is relatively limited, and it faces problems of timeliness and authenticity. To play a role in practical applications, it usually needs to rely on external knowledge sources. The latest research tries to use Retrieval-Augmented Generation (RAG), which combines real-time knowledge sources or specific domain knowledge sources with the implicit knowledge of LLM to complete reasoning or question-answering tasks(Huang & Huang, 2024; Gao et al., 2023; Sawarkar et al., 2024). The naive RAG method achieves this by dividing the text in the document into chunks and mapping these chunks to vector space to calculate the similarity with the query vector. However, its recall often stays at the level of surface semantic similarity, and it is not up to the task when facing knowledge-intensive tasks or complex reasoning tasks.

### 2.2 REASONING OVER KGS

Despite their outstanding performance in various natural language processing tasks, LLMs still have limitations in solving complex tasks based solely on their parameter knowledge, such as multi-hop and knowledge-intensive reasoning. Knowledge graphs store a large number of knowledge triples in the form of graph structures and are widely used to provide LLM with external knowledge supplements. The primary purpose of question answering based on KGs is to retrieve evidence subgraphs from KGs to answer questions, primarily divided into two major categories: semantic parsing and information retrieval.

Semantic parsing analyzes natural language and translates it into representations that a knowledge base can understand to perform reasoning and queries and derive answers. Traditional methods mainly generate query graphs through multiple steps such as entity linking, attribute recognition, and constraint mounting, or use Encoder-Decoder models to transform Semantic Parser problems into Seq2Seq problems through tree decoders(Zhang et al., 2023; Xie et al., 2022; Ye et al., 2021). However, these methods rely on a large amount of annotated data, and their transferability is limited. Some other works(Li et al., 2023) attempt to use the context learning ability of LLM to use the context learning ability of LLM to generate the graph query language corresponding to the problem directly. However, they overly rely on the understanding capabilities of LLMs, and are not proficient in handling complex reasoning problems.

Traditional information retrieval methods use neural networks to identify key entities in queries and connect with KGs to extract candidate answers. This method reduces the use of manual templates, but the model has poor interpretability and mediocre performance. To this end, many works attempt to utilize the powerful thinking ability of LLM to gradually retrieve and generate reliable and

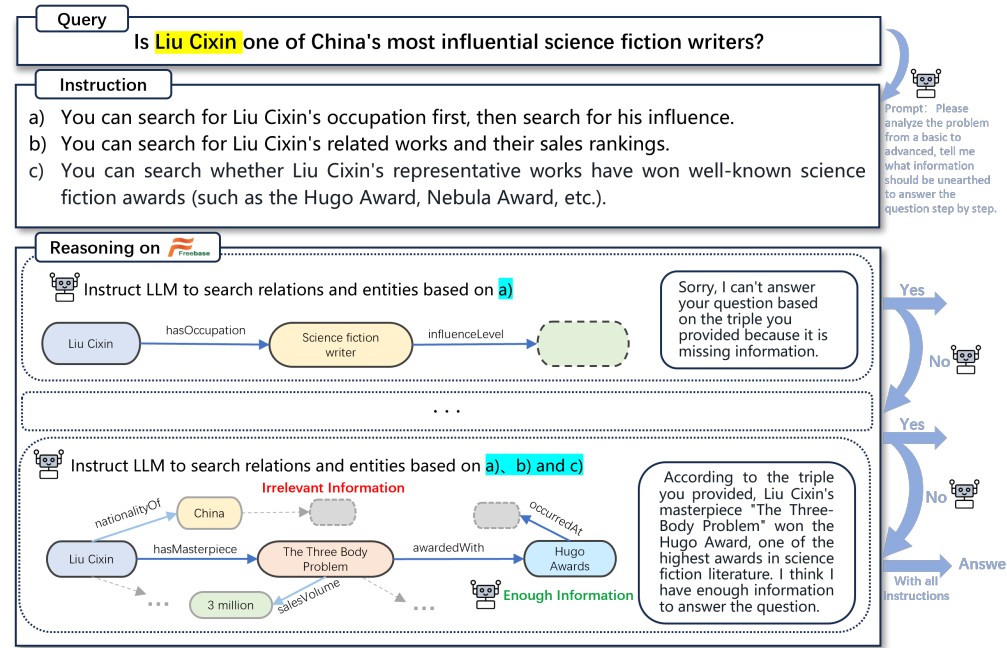

Figure 2: Workflow diagram of ToG-I, where darker relationships represent higher scoring reasoning paths.

interpretable evidence subgraphs along the way. Sun et al. (2024) leveraged the natural language processing capabilities of LLMs, designed an information interaction mechanism between KGs and LLMs, iteratively reasoning to find solutions to problems step by step. Wen et al. (2023) integrate multi-hop reasoning and multi-path exploration, significantly improving the accuracy, transparency, and interpretability of question answering. Liu et al. (2024) realize generalization across different KGs through pseudo graph generation and atomic knowledge verification, and successfully apply it to open-ended question answering. Unlike our method, these models focus on how to retrieve the correct knowledge, while ignoring the importance of uncovering the underlying logic and potential intent of the question.

## 3 METHODS

The main framework of ToG-I is shown in Figure 2. We first prompt the LLM to analyze the potential intent and underlying logic of the question to generate instructions. Then, under the guidance of the instructions, we iteratively explore various possible reasoning paths on the KGs until the LLM determines that the question can be answered based on the current reasoning path. We divide it into two main parts: Instruction Generation and Instruction-based Graph Exploration, and introduce them in detail respectively.

### 3.1 INSTRUCTION GENERATION

Given a question $Q$, with the help of the powerful natural language understanding ability of LLMs, the question is deeply analyzed from multiple angles and levels, and different guiding reasoning ideas $I = \{i_1, i_2, \ldots, i_n\}$ for this question are obtained from shallow to deep. And it is suggested to extract the subject entity $E^0 = \{e_1^0, e_2^0, \ldots, e_m^0\}$ that is most likely to answer the question from different reasoning ideas $I$ and question $Q$ as the starting point of path reasoning $P$.

For example, for the question "Who are the most influential science fiction writers?" When we prompt LLM to analyze the question, we may get guiding opinions such as "You can look for authors with the highest sales of novels in recent years", "You can look for authors who have recently won

science fiction awards (such as the Hugo Awards)", etc. In this way, we can extract "science fiction writers", "Hugo Awards" and other subject entities from it.

## 3.2 Instruction-based Graph Exploration

Next, we will prompt the LLM to perform reasoning through two levels of iteration: outer instructions integration and inner reasoning exploration.

**Instructions integration:** This step involves traversing all the sets of guiding opinions $I = \{i_1, i_2, \ldots i_n\}$. These principles are derived from the LLM's internal knowledge base and are sequenced from superficial to in-depth, facilitating the reasoning process. To make full use of these reasoning opinions, we will summarize the current accumulated set of guiding opinions $I_T = \{i_1, i_2, \ldots i_T\}$ during the $T$-th directive iteration, forming a new, more comprehensive set of guiding opinions $I'_T$. This gradually expands the scope of the directive to achieve dynamic exploration of difficult and easy questions.

**Reasoning exploration:** In the inner iteration, we will explore the reasoning path P, starting from the entity $E^0_{i_t}$, according to the guiding opinions $I'_T$. Specifically, in the $D + 1$-th exploration iteration, we take the set $E^D = \{e^D_1, e^D_2, \ldots e^D_k\}$ composed of all tail entities in the currently explored path set P as the starting point, and explore its top-$K$ associated relationships $R^D = \{r^D_1, r^D_2, \ldots r^D_k\}$ and the corresponding next-level tail entities $E^{D+1} = \{e^{D+1}_1, e^{D+1}_2, \ldots e^{D+1}_k\}$. In this way, after the $D + 1$-th exploration iteration, each path in the path set $P$ contains $D$ evidence triples, after each new triple is obtained, the LLM is prompted to evaluate whether the current reasoning path is sufficient to generate an answer. This mainly includes three stages: instructions-based relationship exploration, instructions-based entity exploration, and answer verification.

- **Instructions-based Relationship Exploration:** In the relationship exploration phase, we adopted a progressive exploration strategy to gradually explore the relationships involved by the entity set $E^0 = \{e1, e2, \ldots em\}$. Specifically, we focus on all incoming and outgoing relationships associated with these entities. With the help of the guiding opinion $I'_T$, we guide the LLM to filter out W potential key relationships $R_m$ from these relationships and score them. It's worth noting that we did not fix the value of $W$, but let $W = T$, meaning that as the guiding opinions accumulate, we will explore more relationships to adapt to the increasingly deep reasoning needs. This method not only maintains the efficiency of the reasoning framework, but also can be flexibly adjusted according to the difficulty of the problem. For simple questions, such as "Who influenced xx", the model can quickly lock the key relationship "influenced_by" based on the guiding opinions, thus avoiding unnecessary calculations and improving reasoning efficiency. For complex multi-hop questions, the framework can gradually broaden the search range, deeply explore potential relationships, and ensure that accurate answers are found within an appropriate range. This dynamic adjustment strategy, based on real-time evaluation of the complexity of the problem, helps to optimize the reasoning path and improve the generalization ability of the model.

- **Instructions-based Entity Exploration:** In the entity exploration phase, we maintained the same idea as in the relationship exploration. Based on the w relationships obtained from the relationship exploration, we first obtain all entities corresponding to different relationships, and under the guidance of the guiding opinion $I'_T$, we select the W entities $E_m$ that are most likely to answer the question from them, also let W=T, and score the selected entities. Then calculate the triple score, that is, for the entities $E_m$ and their corresponding relationships $R_m$, calculate the product of the entity and relationship scores, and select the t triples with the highest scores, so we get another evidence triple on the reasoning path.

- **Answer Verification:** After the reasoning path $P$ is linked to a new triple, we prompt the LLM to judge whether it can answer the question based on the existing reasoning path. If the LLM thinks it can answer the question, stop all iterations and prompt the model to answer the question based on the reasoning path. Otherwise, repeat the steps of integrating guiding ideas and exploration until all guiding opinions are integrated and the exploration reaches the maximum depth. If the LLM still thinks it cannot answer the question at this time, use the inherent knowledge of the LLM to generate an answer.

# 4 EXPERIMENTS

## 4.1 EXPERIMENTAL SETUP

**Datasets:** We evaluated the ability of Tog-I to handle knowledge-intensive tasks on three multi-hop Knowledge Graph Question Answering(KBQA) datasets, including WebQSP(Yih et al., 2016), CWQ(Talmor & Berant, 2018), and GrailQA(Gu et al., 2021), which contain up to four-hop questions. To save computational cost, we randomly selected 1000 samples from each of the three datasets for testing. We used Freebase(Bollacker et al., 2008) as the data source. Freebase is a large, multi-domain KG dataset created by Google, which collects a large amount of entity, attribute, and relationship information. It contains more than 250 million entities, each identified by a unique ID and connected to other entities through thousands of relationships. These relationships can be a person's occupation, a country's capital, a movie's director, and so on. Entities and relationships have one or more attributes to describe their features and properties, such as a person's date of birth, a country's area, a movie's release date, etc. For all datasets, we use Hit@1 as the evaluation metric.

**Detail:** In our experiments, we used ChatGPT, GPT-4[1], and Llama2-7B-Chat(Touvron et al., 2023) as the base LLMs respectively. During exploration, the temperature parameter is set to a higher 0.6 compared to ToG to accommodate more diverse guiding opinions. During inference, the temperature parameter is set to 0 to ensure the accuracy of the inference. The maximum token length limit for generation is 256. In all experiments, we set the number and depth D of generated guiding opinions to 3. The beam search's width $W$ increases with the inference depth (from 1 to 3).

## 4.2 BASELINES

We compared ToG-I with widely used baselines and state-of-the-art methods, which are mainly divided into three categories:

1) Question answering based on the LLM's own capabilities:

- **IO**(Brown, 2020)**:**Standard input-output prompts are used for direct input-output testing of the model.

- **Chain of Thougnt (CoT)**(Wei et al., 2022)**:**The model is encouraged to enhance its reasoning ability by generating a series of intermediate reasoning steps.

- **Self-Consistency (SC)**(Wang et al., 2023b)**:**The answer is obtained by multiple sampling iterations and voting.

2) Semantic parsing methods based on traditional or LLM:

- **Rng-kbqa:**(Ye et al., 2021)**:**Rng-kbqa is a knowledge base question answering method that combines ranking and generation techniques, improving performance through iteratively trained rankers and T5-based generators, especially adept at handling unseen KB pattern problems.

- **KB-BINDER:**(Li et al., 2023)**:**KB-Binder method generates drafts of logical forms using LLM, and combines with the knowledge base for entity and relationship binding, achieving few-shot context learning without training, effectively solving the entity and relationship matching problem in knowledge base question answering.

3) Information retrieval methods based on LLM:

- **Rng-kbqa:**(Sun et al., 2024)**:**Think-on-Graph technique allows for tight coupling interaction between LLMs and KGs, driving the LLM agent to step-by-step search and infer the optimal answer on the associated entities of the KG. This achieves traceability, error correction, and modification of knowledge.

---

[1]ChatGPT and GPT-4 is both from https://openai.com/

Table 1: The ToG-I results for different datasets.

| Method | WebQSP | CWQ | GrailQA |
|---|---|---|---|
| *LLM only* | | | |
| IO/ChatGPT | 63.3 | 37.8 | 29.6 |
| CoT/ChatGPT | 61.8 | 38.2 | 28.1 |
| SC/ChatGPT | 61.2 | 40.1 | 29.8 |
| *Semantic Parsing* | | | |
| KB-BINDER | 74.4 | - | 58.5 |
| Rng-kbqa | 76.2 | - | - |
| *Information Retrieval* | | | |
| ToG/ChatGPT | 76.4 | 58.9 | 68.7 |
| ToG/GPT-4 | 82.6 | 72.5 | 81.4 |
| ToG-I/ChatGPT | 78.3 | 61.2 | 70.2 |
| ToG-I/GPT-4 | **83.9** | **74.2** | **82.6** |

## 4.3 MAIN RESULT

As shown in Table 1, our method achieved the best performance on all three datasets. First, compared to using only the knowledge of the large model itself to answer questions, ToG-I achieved a significant improvement on all three datasets by retrieving external knowledge. This result highlights the importance of introducing external knowledge to alleviate the hallucination of LLM. The performance of ToG-I based on GPT-4 also surpassed that of traditional or LLM-based semantic parsing methods.

Compared with the current state-of-the-art method ToG, we achieved a performance improvement of 1.6%,2.3% and 1.5% on three datasets respectively by providing LLMs with a clear and progressive guiding strategy. Particularly noteworthy is that the most significant performance improvement achieved by ToG-I is on the more complex multi-hop reasoning task CWQ dataset. This phenomenon indicates that when facing more complex problems, the richness of latent intent information provides more room to demonstrate our method's advantages, further validating our method's effectiveness in handling complex problems.

## 4.4 ABLATION STUDY

We conducted an ablation study to analyze the effectiveness of different modules in our method. First, we examined the impact of different baseline models on the experimental results. According to the data in Table 2, even the smaller Llama2-7B-Chat model could achieve significant performance improvement by adopting our ToG-I method. This finding provides a feasible option for deploying LLMs locally.

Then, we compared the impact of different instruction provision methods on the performance of ToG-I. The results show that gradually integrating instructions in each instruction iteration can achieve better performance than providing a single instruction. This may indicate that a single instruction makes it difficult for the model to reason more deeply from a global perspective. In addition, we also studied the impact of exploration width on performance. It can be seen that increasing the exploration width can steadily improve the model's performance. And when the exploration width is fixed at 3, its effect is almost the same as the method of gradually increasing the width used in our ToG-I. However, our method of gradually increasing the width can dynamically adjust the inference width while maintaining performance, thereby improving inference efficiency. This result also verifies the effectiveness of our progressive exploration strategy.

In addition, to explore the impact of the number of instructions and the number of iterations based on instructions on performance, we conducted experiments under the setting of prompting LLM to generate up to 1-4 instructions. As shown in figure 3, the performance of ToG-I first increases

Table 2: Ablation experiment: The impact of different modules on the performance of ToG-I. NS indicates that in the instruction iteration, the instructions are not integrated, but a single instruction is provided in order each time. $W$ represents the fixed exploration width in all reasoning processes.

| Method | LLM | WebQSP | CWQ | GrailQA |
|--------|-----|--------|-----|---------|
| CoT | Llama-2 | 59.2 | 42.1 | 22.8 |
| ToG-I | Llama-2 | 70.6 | 59.8 | 52.2 |
| ToG-I(NS) | ChatGPT | 74.1 | 58.2 | 68.7 |
| ToG-I(W=1) | ChatGPT | 62.5 | 44.2 | 60.1 |
| ToG-I(W=2) | ChatGPT | 72.2 | 52.3 | 65.3 |
| ToG-I(W=3) | ChatGPT | 78.3 | 61.2 | 70.2 |
| ToG-I | ChatGPT | **78.3** | **61.2** | **70.2** |

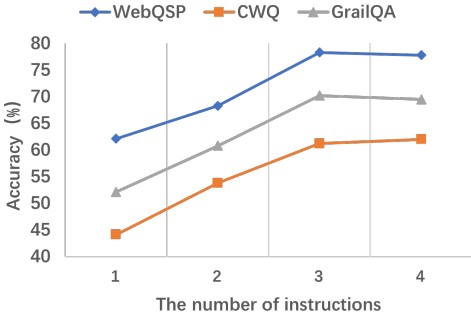 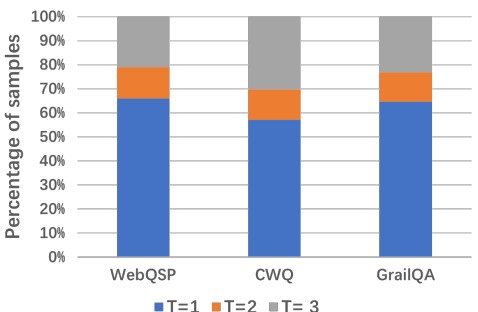

Figure 3: The performance of ToG-I under the instruction of different numbers of instructions.

Figure 4: The position where the question stops iterating in different datasets, where $T$ represents the outer instruction iteration.

and then decreases with instruction increase, reaching the best performance when there are 2 to 3 pieces of instruction. In the experiment, we found that the more instruction the LLM is prompted to generate, the worse the reference of the instruction generated by the LLM becomes, and it may even generate very tricky instruction, which can mislead the LLM's reasoning. Therefore, in all experiments, we prompt the LLM to generate up to 3 pieces of reasoning instructions.

### 4.5 DYNAMIC BEAM SEARCH EXPANSION STRATEGY

To precisely illustrate strategy's effectiveness of gradually expanding the exploration width, we conducted a statistical analysis of the reasoning process on different datasets, paying special attention to the instructive iteration step at the end of reasoning. The data distribution shown in figure 4 indicates that in all test datasets, about 60% of the questions can be answered in the first instructive iteration, and nearly 75% of the questions can be resolved after two instructive iterations. Considering that the number of calls to the LLM increases linearly when using the beam search algorithm, using a fixed maximum beam search width will undoubtedly lead to a large amount of computational resource waste and time consumption. We implemented a strategy of gradually expanding the exploration width to flexibly handle single-hop and multi-hop questions. This method not only ensures the stability of overall performance but also significantly reduces the number of calls to the LLM. Among them, on the WebQSP dataset, our strategy reduced the number of LLM calls by up to about 26%, and about 23% and 18% on GrailQA and CWQ respectively, effectively reducing the consumption of reasoning resources and improving the efficiency of reasoning.

### 5 CONCLUSION

We propose a progressive instructed reasoning framework, ToG-I, which generates guided reasoning ideas and extracts key topic entities to initiate reasoning paths through multi-angle, multi-level anal-

ysis of the problem. Then, under the guidance of these ideas, the framework iteratively explores the KG and dynamically adjusts the exploration range according to the depth of reasoning. The results show that ToG-I outperforms existing methods without increasing training costs and has advantages in knowledge-intensive tasks.

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
