# OpenReview forum: "ToG-I: Progressively Instructed Knowledge Graph-based Large Language Model Reasoning"
_ICLR.cc/2025/Conference — ICLR 2025 Conference Withdrawn Submission_

### Official Review · Reviewer_QA1d · 2024-10-27

**Soundness:** 2
**Presentation:** 2
**Contribution:** 2
**Rating:** 3
**Confidence:** 5

**Summary:**

This paper proposes a new framework called ToG-I that enhances large language models' (LLMs) reasoning by integrating structured knowledge from knowledge graphs (KGs). ToG-I addresses the limitations of current KG-based reasoning methods, which often neglect implicit knowledge and multi-level reasoning, by implementing a progressively instructed approach. This approach guides the LLM through multiple reasoning paths and incorporates commonsense knowledge at various levels, enabling the model to explore different potential answers. The framework dynamically adjusts exploration depth based on task complexity, thus optimizing resource usage. The experiments indicate that ToG-I achieves superior performance in complex reasoning tasks across various datasets.

**Strengths:**

ToG-I’s progressively instructed approach is novel and allows for flexible, multi-level reasoning, addressing complex and ambiguous questions effectively by exploring both explicit and implicit knowledge.

By dynamically adjusting the reasoning depth, ToG-I effectively reduces unnecessary resource consumption while achieving state-of-the-art performance on knowledge-intensive tasks.

The authors tested ToG-I on several datasets, demonstrating its superiority over existing methods, particularly in multi-hop reasoning and complex knowledge tasks. Ablation studies and beam search expansion analysis add rigor to the results.

The framework’s strategy to incrementally increase exploration depth based on task complexity optimizes computational resources, which is practical for real-world applications where efficiency is critical.

**Weaknesses:**

1. For me, this paper is just a simple upgraded version of ToG: https://arxiv.org/pdf/2307.07697, that is, in the first step, the query is decomposed into multiple instructions, and then the same search as ToG. The novelty of the whole paper is relatively low.

2. Many important experiments are missing. In my opinion, this paper is more like an initial version, and many experiments are not shown. For example, experiments using different kg data sources as the retrieval data source; experiments using different triple representation methods (sentences, chains); experiments using different tools to judge similarity; experiments using different search schemes; experiments with different shot numbers in the demonstration; lack of analysis of the results, etc.

3. There are many details missing, which makes it impossible to replicate the experiment. For example, the specific prompts used in each step are not reflected in the paper. (I think the author only wrote a little over 9 pages, and there is enough space or even an appendix to put these contents.)

**Questions:**

1. There are two RNG-KBQA in 2 and 3 of baselines in 4.2. Could the author please explain the similarities or differences between these two, or is this just a typo error?

2. Please use vector graphics in your paper as much as possible.

---

### Official Review · Reviewer_RoVb · 2024-10-31

**Soundness:** 2
**Presentation:** 2
**Contribution:** 2
**Rating:** 3
**Confidence:** 4

**Summary:**

This paper proposes a progressive instructed reasoning framework, ToG-I. Specifically, the proposed framework identifies the core elements, discerns latent intentions, and integrates necessary commonsense reasoning by analyzing the problem from multiple perspectives and levels. Then, ToG-I transforms these analysis results into specific reasoning instructions to guide the LLMs to carry out a progressive reasoning process from a global perspective. Extensive results demonstrate the effectiveness of the proposed method.

**Strengths:**

1. The framework for this work is interesting as it first prompts LLM to analyze the question and then guides LLM to carry out a progressive reasoning process.
2. The integration of knowledge graphs and large language models is a crucial research area.

**Weaknesses:**

1.	The novelty behind the proposed method may be limited as it is primarily built upon existing approaches. The proposed method consists of two main components: prompt LLM to analyze the question (similar to the task decomposition) and then explore the reasoning paths on the KGs (similar to ToG), which are largely derived from existing literature. The exploration stage seems similar to ToG [1], which guides LLM to perform beam search on the KG and iteratively explore the reasoning paths. It would be beneficial to elaborate on the similarities and differences compared to ToG.
2.	There are some mistakes that should be corrected. For instance, in line 318, the name for type 3 should be ToG instead of Rng-kbqa.
3.	The proposed method could be categorized into agent-based method, which utilizes the LLM to analyze the question and searches on the KGs. It would be beneficial to include additional retrieved-based baseline methods for comparison, such as RoG [2], and GNN-RAG [3], which utilize an additional retriever to retrieve relevant facts from KGs and derive new SOTA for this task. This would provide a more comprehensive evaluation of the proposed method.
4.	Settings for just selecting 1000 examples may not be entirely justified. Could you explain your rationale for using the subset of the data and discuss the potential limitations? It would be better to conduct experiments on the whole dataset, consistent with previous methods, to ensure a fair comparison.
5.	It would be beneficial to include analysis experiments for tokens taken to answer the question.

[1] Sun, Jiashuo, et al. "Think-on-Graph: Deep and Responsible Reasoning of Large Language Model on Knowledge Graph." The Twelfth International Conference on Learning Representations.

[2] LUO, LINHAO, et al. "Reasoning on Graphs: Faithful and Interpretable Large Language Model Reasoning." The Twelfth International Conference on Learning Representations.

[3] Mavromatis, Costas, and George Karypis. "GNN-RAG: Graph Neural Retrieval for Large Language Model Reasoning." arXiv preprint arXiv:2405.20139 (2024).

**Questions:**

Please see **Weaknesses** above.

---

### Official Review · Reviewer_5a6g · 2024-11-04

**Soundness:** 2
**Presentation:** 3
**Contribution:** 2
**Rating:** 5
**Confidence:** 4

**Summary:**

This paper introduces ToG-I, which improves existing Think-on-Graph. The existing ToG framework performs beam-search on knowledge graphs with the guidance of LLMs. The ToG-I further improves this framework by prompting LLM to generate multiple (<=3) instructions as guiding opinions to help reasoning path search on KG, and iteratively incorporate the instructions during search to gradually expand the scope to dynamically answer questions of varying difficulties.

The ToG-I achieves SOTA performance overall on three KBQA datasets comparing with baselines and methods in three categories.

**Strengths:**

The writing is clear and easy-to-follow.

The effectiveness of different modules is clearly discussed in the ablation study.

The method is a streamlined extension of previous ToG and demonstrates clear improvements in performance.

**Weaknesses:**

1.	By iteratively following the instructions, the computational cost might be even higher than ToG, which already involves costly beam-search and multiple LLM calls.
2.	This paper claims that the instruction generation helps alleviate the problem of one-sided understanding of the question by LLMs (line 94). However, this problem may not be as apparent in the datasets used for experimentation. The example illustrated in the figures and discussed throughout the paper may naturally exhibit this issue, as it involves subjective questions with answers that can vary based on interpretation. In contrast, one-hop and multi-hop questions experimented with typically have more objective answers. Thus, it is not quite clear if the instruction generation module is the key contributor to the performance gain; or, simply run ToG multiple times and get the final result with, say, self-consistency, would yield comparable improvements.

**Questions:**

1.	Typo: The information retrieval methods based on LLM on line 318 should be ToG.
2.	Could you further explain the missing scores in Table 1? Additionally, description of the datasets used in experiments might be helpful.
3.	The example illustrated in the figure does not appear to be quite representative. Using examples that are similar to those in the experiments would be more effective.
4.	Case study about instances where ToG-I succeeds while ToG fails would be helpful to demonstrate the superiority of ToG-I.

---

### Note · Authors · 2025-01-22

**Comment:**

I have read and agree with the venue's withdrawal policy on behalf of myself and my co-authors.

**Withdrawal Confirmation:**

I have read and agree with the venue's withdrawal policy on behalf of myself and my co-authors.